# Pyrethroid Resistance Situation across Different Eco-Epidemiological Settings in Cameroon

**DOI:** 10.3390/molecules27196343

**Published:** 2022-09-26

**Authors:** Nelly Armanda Kala-Chouakeu, Paulette Ndjeunia-Mbiakop, Idriss Nasser Ngangue-Siewe, Konstantinos Mavridis, Vasileia Balabanidou, Roland Bamou, Mabu Maxim Bindamu, Abdou Talipouo, Landre Djamouko-Djonkam, Jean Arthur Mbida-Mbida, Jeanette Tombi, John Vontas, Timoléon Tchuinkam, Christophe Antonio-Nkondjio

**Affiliations:** 1Vector-Borne Diseases Laboratory of the Applied Biology and Ecology Research Unit (VBID-URBEA), Department of Animal Biology, Faculty of Science of the University of Dschang, Dschang P.O. Box 067, Cameroon; 2Laboratoire de Recherche sur le Paludisme, Organisation de Coordination pour la lutte Contre les Endémies en Afrique Centrale (OCEAC), Yaoundé B.P. 288, Cameroon; 3Faculty of Sciences, University of Yaoundé I, Yaoundé P.O. Box 337, Cameroon; 4Laboratory of Animal Biology and Physiology, University of Douala, Douala P.O. Box 24157, Cameroon; 5Pesticide Science Laboratory, Department of Crop Science, Agricultural University of Athens, 11855 Athens, Greece; 6Research Laboratory of Biochemestry of University of Bamenda, Bambili P.O. Box 39, Cameroon; 7Institute of Molecular Biology and Biotechnology, Foundation for Research and Technology-Hellas, 70013 Heraklion, Greece

**Keywords:** pyrethroid resistance, detoxification genes, eco-epidemiological settings, anopheles, Cameroon

## Abstract

Rapid emergence and spread of pyrethroid resistance in *Anopheles gambiae* populations is among the main factors affecting malaria vector control in Cameroon, but there is still not enough data on the exact pyrethroid resistance status across Cameroon. The present study assessed pyrethroid resistance profile in different eco-epidemiological settings in Cameroon. Susceptibility bioassay tests were performed with F0 *An. gambiae* females aged three to five days. Mosquito susceptibility to both permethrin and deltamethrin was assessed. Species of the *An. gambiae* s.l. complex were identified using molecular diagnostic tools. Target site mutations conferring resistance were detected using Taqman assays. Quantitative reverse transcription-real-time PCR (qRT-PCR) 3-plex TaqMan^®^ assays were used for the quantification of detoxification genes implicated in pyrethroid resistance. *An. gambiae*, *An. coluzzii* and *An. arabiensis* were identified in the different settings. *An. gambiae* was dominant in Santchou, Kékem, Bélabo, Bertoua and Njombé, while *An. coluzzii* was abundant in Tibati and Kaélé. High frequencies of the kdr L1014F allele ranging from 43% to 100% were recorded in almost all sites. The L1014S kdr allele was detected at low frequency (4.10–10%) only in mosquito populations from Njombé and Tibati. The N1575Y mutation was recorded in Kaélé, Santchou, Tibati and Bertoua with a frequency varying from 2.10% to 11.70%. Six Cytochrome P450 genes (*Cyp6p3*, *Cyp6m2*, *Cyp9k1*, *Cyp6p4*, *Cyp6z1*, and *Cyp4g16*) were found to be overexpressed in at least one population. Analysis of cuticular hydrocarbon lipids indicated a significant increase in CHC content in mosquito populations from Kaélé and Njombé compared to Kékem, Bélabo and Bertoua populations. The study indicated high pyrethroid resistance across different ecological settings in Cameroon with different profile of resistance across the country. The present situation calls for further actions in order to mitigate the impact of insecticide resistance on vector control measures.

## 1. Introduction

Malaria is among the most dangerous vector borne diseases in the world. The disease is responsible yearly for 627,000 deaths [1]. Long-lasting insecticidal nets (LLINs) and indoor residual spraying (IRS) are the main interventions recommended by the World Health Organization (WHO) for disease prevention [2]. Pyrethroid insecticides are used for LLINs and IRS because of their high effectiveness and strong excito-repellency effect on insects, and low toxicity to humans and mammals at recommended dosage. Other insecticides, including organophosphates, carbamates and organochlorines are mainly used for indoor residual spraying. These vector control interventions have contributed to a significant decline of malaria prevalence over recent years [2]. However, this progress is now threatened by the rapid expansion of pyrethroid resistance in vector populations [3].

In Cameroon, LLINs are the main tools used for malaria vector control. Four large-scale distribution campaigns of bed nets have so far been conducted in the country. Over 60% of households are considered to possess LLINs [4]. A recent review on insecticide resistance evolution between 1990 and 2017 indicated that pyrethroid resistance has rapidly evolved in most epidemiological settings [5]. It is possible that the increasing use of LLINs may have contributed to increase insecticide selective pressure on mosquito populations. Moreover, the uncontrolled use of pesticides in agriculture is also considered to contribute to vector resistance to pyrethroids [5,6,7].

Two main mechanisms are involved in pyrethroid resistance in *An. gambiae* s.l., target site resistance and metabolic resistance [8]. Amino acid substitutions, which lead to resistance to pyrethroids or DDT, are commonly referred to as knock-down resistance (kdr) mutations. The most well-known involve the replacement of the amino acid leucine on codon 1014 by phenylalanine (L1014F or “*kdr-west*” mutation) or by serine (L1014S or “*kdr-east*” mutation) [9]. Another mutation, known as N1575Y due to the substitution of asparagine by tyrosine at position 1575 of the domain III-IV of VGSC of *An. gambiae* has also been reported in pyrethroid resistant *An. gambiae* [10]. This mutation is considered to increase the resistance status of samples displaying the L1014 mutation [11,12,13,14]. Metabolic resistance is due to the overexpression of enzymes capable of detoxifying or sequestering insecticides. Three main enzymes families are considered to confer metabolic based resistance; these include esterases, glutathione S transferase and P450 monooxygenases. The overexpression of cytochrome P450 is the main mechanism associated with malaria vector resistance to pyrethroids. Several genes, including *Cyp6p3, Cyp6m2, Cyp6z2, Cyp9k1, Cyp6p9a, Cyp6p9b* and *Cyp6z1* have been reported to confer pyrethroid resistance in *An. gambiae* s.l. or *An. funestus* [12,13,15,16,17,18,19,20,21,22].

Although in recent years there have been an increasing number of studies assessing pyrethroid resistance levels in malaria vectors in Cameroon [7,8,15,16,23,24,25,26,27,28,29], there is still not enough data providing a global picture of the pyrethroid resistance situation in different eco-epidemiological settings. The present study aimed to provide updated data on pyrethroid resistance profiles in *An. gambiae* s.l. population distributed in different eco-epidemiological settings (Sahelian, Sahelo-Soudanese, highland, coastal and forest) across Cameroon.

## 2. Methods

### 2.1. Study Sites

The study was conducted in five eco-epidemiological settings in Cameroon: the Sahelian zone (Kaélé), the Sahelo-Soudanese zone (Tibati), the forest zone (Bertoua and Bélabo), the western highlands (Kékem and Santchou) and the littoral zone (Njombé) (Figure 1). Characteristics of each study site are described in detail in Kala-Chouakeu et al. [30] and Ngangue-Siewe et al. [31] (see Appendix A). The study was carried out during three consecutive years in July (2019, 2020, 2021) in Bertoua and Santchou, in July 2020 and July 2021 in Kaele and Tibati and July 2021 in Kékem Njombé and Bélabo (cross-sectional survey) (Figure 1).

### 2.2. Mosquito Larvae Collection

Larvae and pupae were collected in the different studied areas using the standard dipping method [32] in various breeding sites (ponds, roadside ditches, marshes, shallow wells and river banks). In each study site, collected larvae were pooled together and reared under controlled conditions (temperature 27 ± 2 °C, relative humidity 70 ± 10%) with Tetramin^®^ baby fish food until adult’s emergence. Emerged adults (F0) were identified using morphological identification keys [33] and were used for susceptibility tests.

### 2.3. Susceptibility Bioassays

Bioassays were carried out using the standard WHO protocol [34]. Tests were performed with the WHO supplied insecticide-impregnated papers. Insecticides tested included two pyrethroids at different doses (0.05% deltamethrin (1×, 0.25% deltamethrin (5×), 0.5% deltamethrin (10×),0.75% permethrin (1×), 3.75% permethrin (5×) and 7.5% permethrin (10×). Three- to five day-old unfed *An. gambiae* s.l. females were left for observation for one hour before they were exposed for one hour to insecticides. Experiments were conducted at a temperature of 22 to 26 °C with a minimum of four replicates per bioassay and the mortality rates was recorded after 24 h. The insecticide susceptible strains *An. gambiae* Kisumu and Ngousso strains were used as control to assess the quality of the impregnated papers. For control tests, silicone-treated papers were used. Susceptibility is indicated by a mortality > 97%. If the observed mortality rate is between 90 and 97%, the presence of resistant genes in the vector population must be confirmed through additional bioassay tests and/or molecular assays. If mortality is less than 90%, the population is considered resistant [11]. For each insecticide, dead mosquitoes were kept separately in 1.5-mL microtubes containing 70% ethanol, whereas mosquitoes that were still alive after the tests and control samples were kept separately in RNA later tubes for molecular analysis. In each location, a subset of 30 to 40 unexposed 3- to 5-day-old non-blood fed female *An. gambiae* s.l. were stored in RNA later for characterization of molecular mechanisms (species ID, knockdown resistance and overexpression of detoxification resistance genes).

### 2.4. Synergist Bioassay with Piperonyl Butoxide (PBO)

The PBO synergist was used to assess the potential contribution of P450 monooxygenase enzymes. For this purpose, subsamples of 20 to 25 unfed, 3- to 5-day-old adult females *An. gambiae* s.l. randomly collected from a cage were pre-exposed to 4% PBO paper for 1 h before being immediately exposed to 0.75% permethrin or 0.05% deltamethrin for an additional one hour. Mortality following exposure to both PBO and permethrin or deltamethrin was recorded after 24 h. These tests were conducted alongside controls.

### 2.5. Mosquito Processing

#### 2.5.1. Total RNA and DNA Extraction from Mosquito Pools

A magnetic bead-based approach with the MagSi kit was used to extract total RNA and DNA from mosquitoes (MagnaMedics Diagnostics B.V., De Asselen Kuil 12, 6565 RD Geleen, Netherlands). The quantity and purity of total DNA and RNA were assessed spectrophotometrically using Nanodrop measurements. RNA quality was assessed by 1.0% *w/v* agarose gel electrophoresis.

#### 2.5.2. Genotyping of Mosquito Samples and Multiplex RT-qPCR for Gene Expression Analysis

Species identification and target site mutation determination (kdr (L1014F, L1014S) and N1575Y) were performed using the modified IVCC Vector Population Monitoring Tool (VPMT) protocol described by Jones et al. [17]. The analysis of molecular species was based on a newly developed TaqMan assay based on SINE200 retrotransposon insertion polymorphisms within the speciation islands of *Anopheles gambiae* molecular species [30,32]. Allele frequency was calculated with regression models using a methodology developed by Mavridis et al. [20].

The TaqMan^®^ 3-plex quantitative real-time reverse transcription PCR (qRT-PCR) assays were used for the quantification of expression of detoxification genes using RPS7 for normalization purposes in each assay [20].

#### 2.5.3. Cuticular Hydrocarbons (CHCs) Identification and Quantitation by GC-MS and GC-FID

Quantification of CHC was performed in the Molecular Entomology Laboratory of IMBB-FORTH. Female mosquitoes from five populations (Njombé, Kékem Bélabo, Kaélé and Bertoua) were collected. Prior to analysis, the mosquitoes were air-dried at 25 °C and pooled (three replicates of 20 female mosquitoes). Their dry weight was measured and the corresponding samples were analyzed for CHC. The identification and quantification of CHCs (by GC-MS and GC-FID) was performed as previously described by Balabanidou et al. [21], with minor modifications. Cuticular lipids from samples were extracted by a 1-min immersion in hexane with gentle agitation; the extracts were pooled and evaporated under a stream of N2. The CHCs were separated from the other components and finally concentrated before solid phase extraction (SPE) chromatography. Quantitative amounts were estimated by co-injection of n C24 as internal standard (2890 mg/mL in Hexane). CHC quantification was calculated as the sum of the areas of the 32 peaks in total, using the internal standard.

### 2.6. Statistical Analysis

The calculation of fold-changes, 95% CIs and statistical significance was performed according to the Pfaffl method [20]. Graphs were constructed using Excel software. Mortality rate was expressed as the ratio of the number of mosquitoes found dead or unable to stand on their legs to the number exposed. Confidence intervals were calculated using Medcalc 15.8. The comparison of mortality rate and fold change was performed using the chi-squared test and Student’s *t*-test, respectively.

## 3. Results

### 3.1. Resistance Profile

A total of 9813 field *An. gambiae* s.l. mosquitoes were screened to assess their susceptibility level to both permethrin and deltamethrin (Figure 2). These included 2340 mosquitoes from Santchou, 2245 from Bertoua, 1522 from Tibati, 1519 from Kaélé, 800 from Njombé, 640 from Kékem and 747 from Bélabo. The laboratory strains (Kisumu and Ngousso) were found to display a mortality rate of 100% for both permethrin 0.75% and deltamethrin 0.05%. Low mortality rates (<70%) were recorded in all sites with both permethrin 0.75% and deltamethrin 0.05%, except in Njombé, where a mortality rate close to 91% was registered for deltamethrin 0.05% (Figure 2). Temporal variation of the pyrethroid resistance profile was recorded within each site (Appendix A).

#### 3.1.1. Intensity of Permethrin and Deltamethrin Resistance

The mortality rate of *An. gambiae* s.l. was found to increase with the increasing concentration of deltamethrin and permethrin (Figure 3). Low resistance intensity to both permethrin and deltamethrin was recorded in Njombé (mortality rate at 5× > 98%; 95% CI [0.80–1.20]) (Figure 3A). In the remaining sites, *An. gambiae* s.l. was found to display moderate to high resistance intensity to permethrin (Figure 3A). Resistance intensity to deltamethrin was high in almost all sites (mortality rate to 10× concentration < 90%) (Figure 3B). Figure 3 also presents the distribution of *An. gambiae* complex species in the different study sites.

#### 3.1.2. Species ID

A total of 637 specimens were genotyped to determine species present in each site. *An. coluzzii, An. arabiensis* and *An. gambiae* were detected in Kaélé. In Bertoua, Bélabo and Njombé, both *An. gambiae* and *An. coluzzii* were recorded. In Santchou and Kékem, *An. gambiae* was the only species present, whereas in Tibati only *An. coluzzii* was recorded (Figure 3A,B).

#### 3.1.3. Bioassays with the Synergist Piperonyl Butoxide (PBO)

To test whether the increased resistance to both permethrin and deltamethrin is mediated through the action of P450s genes, mosquitoes were pre-exposed to the synergist PBO, a well-known P450 inhibitor, before being exposed to either permethrin or deltamethrin. Pre-exposure to PBO increased mortality to both permethrin 0.75% and deltamethrin 0.05% in all sites (Figure 4). Partial recovery of susceptibility was obtained when mosquitoes were exposed to permethrin supporting the influence of different resistance mechanisms. With deltamethrin, full recovery of susceptibility was observed after pre-exposure to PBO (mortality: 100%) in Njombé and Kaélé whereas partial recovery was recorded in Tibati (75.70%) and Bertoua (75.75%). This suggests that different mechanisms may be interplayed in the different sites.

### 3.2. Target Site Mutations (kdr L1014F/S, kdr N1575Y)

The distribution of different mutations associated with pyrethroid resistance was assessed (Table 1). The frequency of L1014F allele (kdr West) was always >75% except in Kaélé (43.40%). The L1014S kdr allele (kdr East) was detected only in the populations of Njombé and Tibati at low frequencies. The N1575Y allele was recorded in Kaélé, Santchou, Tibati and Bertoua at a frequency varying from 2.10% to 11.70%.

### 3.3. Expression Analysis of Genes Implicated in Insecticide Resistance

Quantitative PCR analyses were conducted to assess the expression profile of eight detoxification genes involved in insecticide resistance. This includes *Cyp6p3*, *Cyp6m2*, *Cyp9k1*, *Cyp6p4*, *Cyp6z1*, *Cyp6p1* and *Cyp4g16* [31]. High overexpression ratios were obtained when field populations were compared to the Kisumu strain (*An. gambiae*) originating from East Africa (Table 2). On the other side, moderate overexpression ratios were obtained when field populations of either *An. gambiae* or *An. coluzzii* were compared to the Ngousso laboratory strain (*An. coluzzii*). The following genes *Cyp9k1*, *Cyp6m2* and *Cyp6p4* displayed the highest overexpression fold-changes in different comparisons (Table 2).

### 3.4. Analysis of Cuticular Hydrocarbon (CHC) Lipids as a Marker of Pyrethroid Resistance

The mean amount of CHCs normalized for dry body weight (ng CHCs/mg dry body weight ± SD) from five Anopheles mosquito populations was calculated as 2609 ± 185 in Njombé, 2152 ± 111 in Kékem, 2150 ± 41 in Bélabo, 2964 ± 114 in Kaélé and 2091 ± 319 in Bertoua (Figure 5). First Kaélé and then Njombé mosquitoes had significantly higher amounts of CHCs compared to the rest of mosquitoes normalized for their size differences. The difference of the means between Kaélé and Njombé, Kaélé and Kékem, Kaélé and Bélabo and Kaélé and Bertoua (*p* = 0.04, *p* = 0.0009, *p* = 0.0003, *p* = 0.01, respectively, as well as Njombé and Kékem, Njombé and Bélabo, Njombé and Kaélé populations was found to be significant (*t*-test, *p* = 0.02, *p* = 0.01 and *p* = 0.04, respectively). It should be noted that their CHC content was also higher when compared to CHC levels of both laboratory strains (Kisumu and Ngousso, 2463 ± 145.08 and 1675 ± 26.08 ng CHCs/mg of dry mosquito weight respectively), but only found significantly higher when compared to the Ngousso strain (*t*-test, *p* = 0.03).

## 4. Discussion

The main objective of the study was to characterize pyrethroid resistance profiles in *An. gambiae* s.l. populations in five eco-epidemiological settings in Cameroon. A high resistance profile of *An. gambiae* s.l. to both deltamethrin and permethrin was recorded in all sites. The resistance profile detected during the present study, was in accordance with studies conducted so far across the country, suggesting rapid expansion of insecticide resistance in vector populations [5,7,35]. Similar evolution of pyrethroid resistance patterns have been reported across Africa [6,18,25]. The high pyrethroid resistance profile detected in vector populations could result from the prevailing insecticide selective pressure on mosquitoes following regular scaling up since 2010 of pyrethroid-treated nets across the country. Different brands of pyrethroid-treated nets with different active compounds, including permethrin, alpha-cypermethrin and deltamethrin, have been deployed in the country [36]. The domestic use of insecticide spray and coils is also largely practiced by the population and could somewhere contribute to the actual trend of resistance recorded across the country [33,34].

Significant variation of the susceptibility level was detected between eco-epidemiological sites and supports a heterogeneous pattern of insecticide resistance in *An. gambiae* s.l. populations across the country. The Njombé population in the littoral region, was the only one fully susceptible to 10× concentration of both permethrin and deltamethrin, whereas in the remaining sites, lower mortality rates to 5× and 10× diagnostic concentration of both permethrin and deltamethrin were recorded, suggesting high resistance intensity in those sites. This resistance profile could affect the performance of pyrethroid-treated nets, yet this deserves further investigation. The populations of Santchou and Kékem in the highland area and Kaélé in the Sahelian region display the highest resistance intensities with lowest mortality rates recorded for 5× and 10× diagnostic concentrations of both permethrin and deltamethrin. It is likely that mosquitoes in the sites of Kaélé, Santchou, Kékem and Bélabo are exposed to additional selective pressure resulting from the use of pesticides in agriculture and or public health. A recent study indicated the use of a large variety of compounds, including herbicides, fungicides and insecticides, in agriculture in Cameroon [7]. Kaélé is situated in an area with intense cultivation of cotton. Previous studies in localities close to the site documented frequent use of pyrethroids, organochlorine and carbamates as pesticides in cotton cultivation [25]. Santchou, Kékem and Bélabo are situated in areas with intensive cultivation of crops such as cocoa, vegetables, plantains which also request frequent use of pesticides.

A high resistance profile to permethrin and deltamethrin was recorded in both *An. gambiae* and *An. coluzzii*. In Cameroon, *An. gambiae* populations are mainly distributed in periurban and rural settings, whereas *An. coluzzii* is predominant in urban centres [36]. It is considered that this distribution might somewhere influence the type of environmental pollutants and selective pressure each species is exposed to [16,36,37,38].

High frequencies of the knock down resistance allele L1014F was recorded. The allele was close to fixation in all sites except in Kaélé. The L1014S mutation was recorded in low frequencies in Njombé and Tibati. These findings are consistent with previous reports indicating the predominance of the L1014F allele across the country [5,12]. The N1575Y mutation, also contributing to DDT and pyrethroid resistance, was detected in Bélabo, Kaélé, Tibati, Bertoua and Santchou populations but at very low frequencies, supporting no major role for this mutation.

Bioassay analysis with PBO as synergist indicated full recovery of susceptibility to deltamethrin in Kaélé and Njombé, whereas in the remaining sites, partial recovery was recorded. These results probably support a major role played by P450 genes in pyrethroid resistance in Kaélé and Njombé, whereas in other sites, other mechanisms could be implicated. Several key P450 genes were found to be overexpressed in at least one population. These include *Cyp6p3*, *Cyp6m2*, *Cyp9k1*, *Cyp6p4*, *Cyp6z1* and *Cyp4g16*. The detoxification gene *Cyp9k1* was the most frequently and extensively upregulated gene in Njombé, Bertoua and Kékem populations. This gene has frequently been associated with resistance to pyrethroid in *An. gambiae* in Cameroon [13]. Functional expression analysis in *An. gambiae* indicated that *Cyp9k1* was able to metabolize deltamethrin, as well as pyriproxyfen [39,40]. *Cyp6p4,*
*Cyp6z1, Cyp6p3 and Cyp6m2* were also found to be upregulated in at least one population. *Cyp6p4, Cyp6m2* and *Cyp6p3* have been shown to metabolize pyrethroids (permethrin, etofenprox, deltamethrin) [36,37,39]. *Cyp6z1* have been reported to metabolize DDT in *An. gambiae* [39]. *Cyp4g16,* which was found to be upregulated in Tibati, is a functional oxidative decarboxylase gene known to catalyze the last step of epicuticular hydrocarbon biosynthesis and could contribute to insecticide resistance via the enriched synthesis of CHCs, thus reducing pyrethroid uptake [41,42,43]. Unfortunately, the limiting amount of mosquito individuals from Tibati hampered the analysis of CHC content from this site. Nevertheless, the CHC analysis of mosquitoes from other sites were found fruitful. Mosquitoes originating from Njombé and especially from Kaélé had higher levels of CHCs compared to mosquitoes from Kékem, Bélabo and Bertoua, as well as from the two laboratory strains Ngousso and Kisumu. The former populations (Njombé and Kaélé) did not overexpress *Cyp4g16*, but it could be that other enzymes, located upstream in the CHC biosynthetic machinery, are differentially expressed and affect the final amounts of the produced CHCs. Although CHCs abundance affect insecticide resistance levels [41,42] it has also been demonstrated that it could influence mating success [44]. CHCs genes alongside metabolic base resistance genes and target site resistance could contribute to the general profile of resistance detected in vector populations in Cameroon.

## 5. Conclusions

The study indicated high profile of insecticide resistance in different eco-epidemiological settings across Cameroon. The present findings highlight the need for the implementation of insecticide resistance management strategies across the country. The replacement of current treated nets by new generation nets treated with different insecticides or mixtures of insecticides should be envisaged. Furthermore, the implementation of integrated control strategies could be critical to address challenges in different ecoepidemiological settings and to improve malaria control in the country.

## Figures and Tables

**Figure 1 molecules-27-06343-f001:**
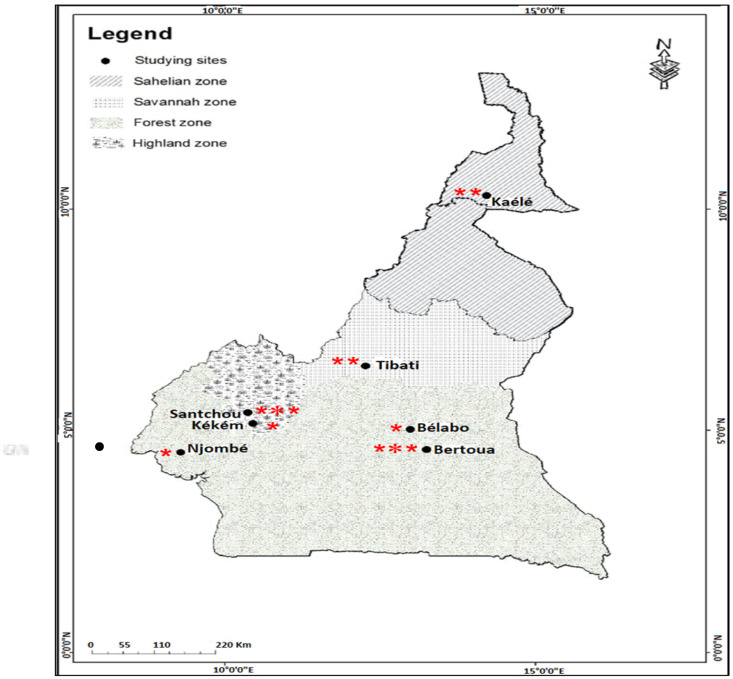
Map showing study sites with the number of collection periods (the number of red star ***** in site denote the number of collection time).

**Figure 2 molecules-27-06343-f002:**
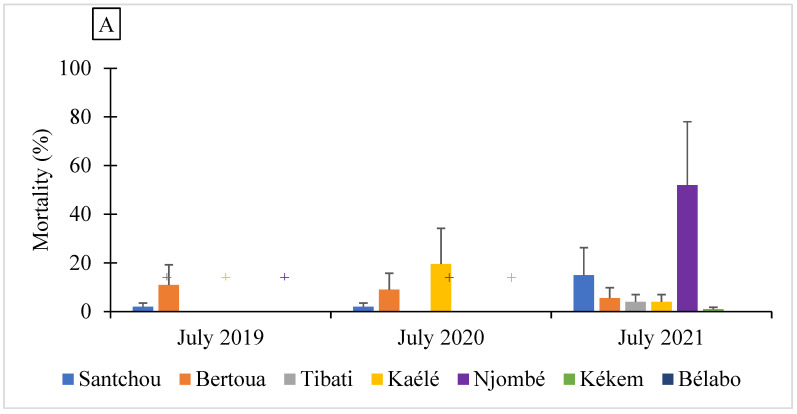
Susceptibility profile of *Anopheles gambiae* s.l. to permethrin 0.75% (**A**) and deltamethrin 0.05% (**B**) (+ no bioassay conducted in the site).

**Figure 3 molecules-27-06343-f003:**
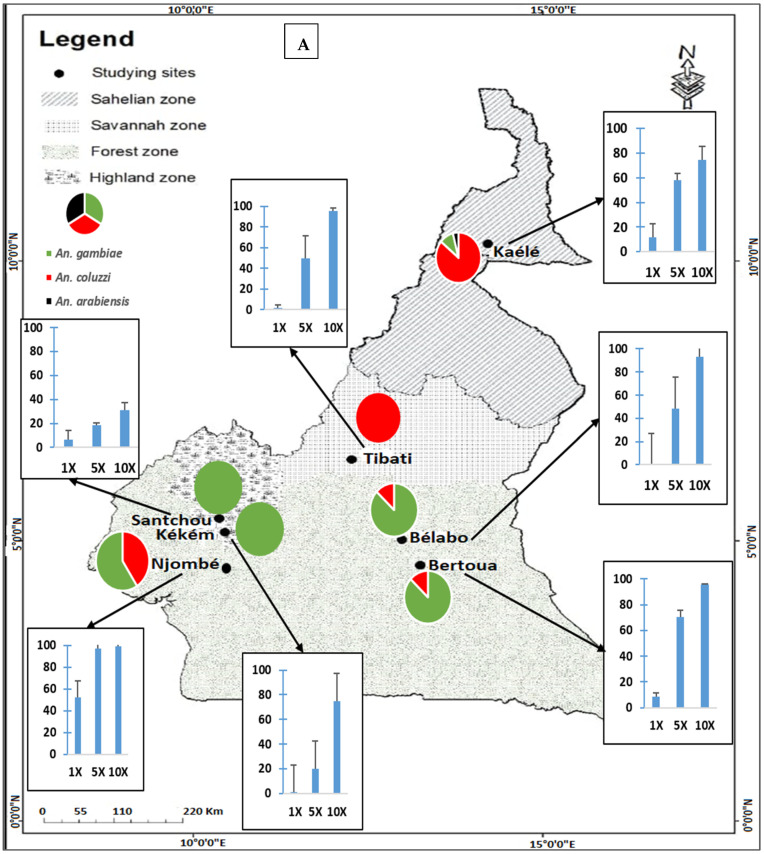
Resistance intensity of *An. gambiae* (s.l.) populations to permethrin (**A**) and deltamethrin (**B**) in the different study sites in Cameroon. (Permethrin (1×, = 0.75%, 5×, = 3.75%, 10×, = 7.5%) and deltamethrin (1×, = 0.05%, 5×, = 0.25%, 10×, = 0.5%)).

**Figure 4 molecules-27-06343-f004:**
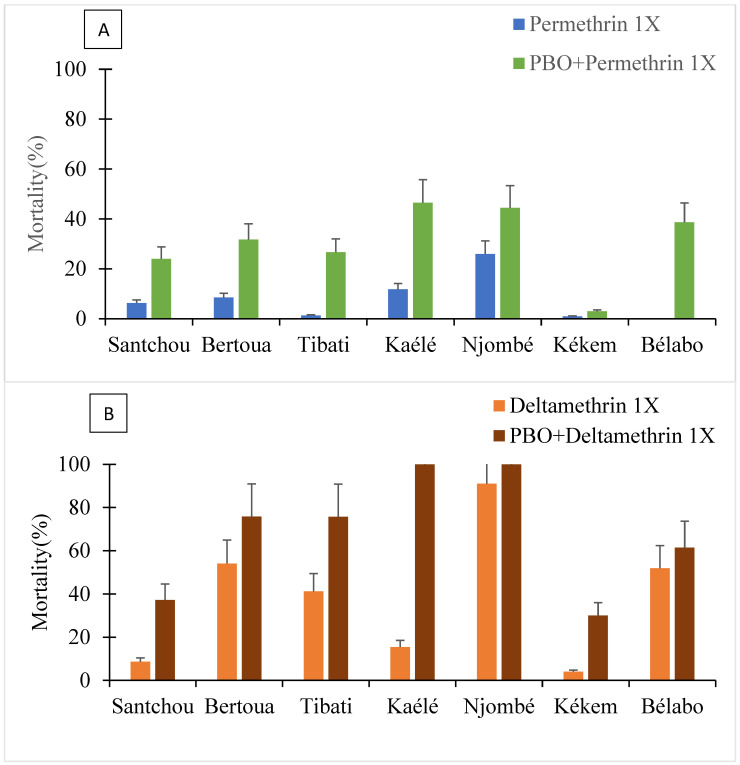
Mortality rate of *An. gambiae* s.l. after synergist assays with permethrin (**A**) and deltamethrin (**B**).

**Figure 5 molecules-27-06343-f005:**
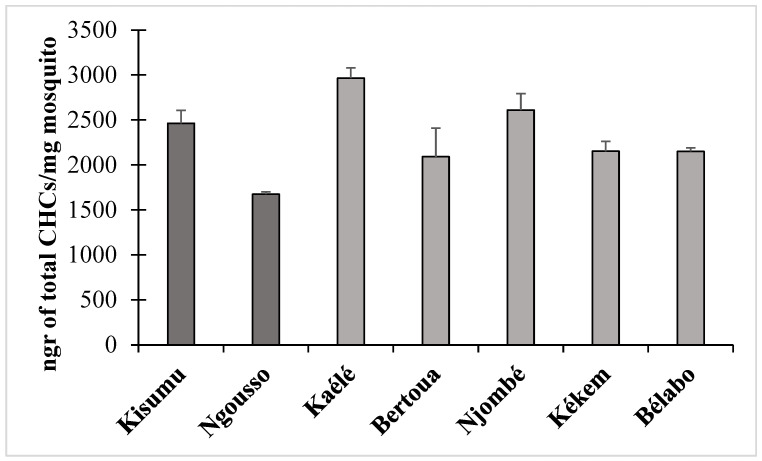
Mean CHC amounts from *Anopheles gambiae* populations from Kaélé, Bertoua, Njombé, Kékem and Bélabo (Cameroon).

**Table 1 molecules-27-06343-t001:** Frequencies of kdr L1014F, L1014S and N1575Y resistance alleles in *An. gambiae* s.l. populations.

Ecological Setting/Strain	Population	Sample Size	Resistant Allele Frequencies (Mean ± SE)		
			% kdr L1014F	% kdr L1014S	N1575Y
Kisumu & Ngousso Susceptible strain	Kisumu lab strain	40	0.0 ± 0.0	0.0 ± 0.0	0.0 ± 0.0
	Ngousso lab strain	40	0.0 ± 0.0	0.0 ± 0.0	0.0 ± 0.0
Sahelian Zone	Kaélé	40	43.4 ± 5.7	0.0 ± 0.0	11.7 ± 4.8
Humid savanah	Tibati	40	79.9 ± 5.1	4.9 ± 0.1	3.75 ± 1.3
Highland zone	Santchou	40	100.0 ± 0.0	0.0 ± 0.0	5.5 ± 4.0
	Kékem	40	100.0 ± 0.0	0.0 ± 0.0	0.0 ± 0.0
Forest zone	Njombé	30	86.1 ± 8.7	10.0 ± 2.1	0.0 ± 0.0
	Bertoua	40	94.6 ± 3.2	0.0 ± 0.0	2.1 ± 0.9
	Bélabo	40	98.8 ± 1.25	0.0 ± 0.0	0.0 ± 0.0

SE: standard error.

**Table 2 molecules-27-06343-t002:** Gene expression analysis in the seven resistant mosquito populations compared to the susceptible strains. Bold letters indicate statistically significant upregulation, and asterisks (*) indicate consistent upregulation compared to both susceptible strains.

Population	Fold Change (95% CI)						
Comparison	*CYP6P3*	*CYP6M2*	*CYP9K1*	*CYP6P4*	*CYP6Z1*	*CYP6P1*	*CYP4G16*
Njombé vs. KIS	2.59	0.490	21.2 *	1.92	2.23 *	1.36	1.53
	(0.746–6.99)	(0.306–0.874)	(9.14–79.8)	(0.690–6.695)	(1.63–3.69)	(0.929–2.33)	(0.831–4.13)
Njombé vs. NG	0.683	0.267	8.14 *	1.29	1.60 *	0.639	1.02
	(0.244–1.65)	(0.140–0.458)	(4.910–13.81)	(0.553–3.789)	(1.22–2.10)	(0.446–1.11)	(0.753–1.31)
Bertoua vs. KIS	2.70	2.16	8.29 *	2.99 *	1.627	1.30	1.40
	(1.44–3.80)	(1.46–3.39)	(3.14–30.9)	(1.70–5.04)	(0.996–3.076)	(0.910–2.09)	(0.564–4.29)
Bertoua vs. NG	0.713	1.18	3.18 *	2.01 *	1.17	0.611	0.928
	(0.545–0.910)	(0.670–1.72)	(1.81–5.09)	(1.37–2.85)	(0.745–1.78)	(0.436–0.993)	(0.511–1.34)
Kaélé vs. KIS	3.8	0.169	3.72	18.2 *	0.981	1.32	0.560
	(2.00–5.52)	(0.117–0.294)	(1.24–15.1)	(6.94–33.9)	(0.768–1.591)	(0.990–1.83)	(0.218–1.91)
Kaélé vs. NG	1.02	0.092	1.43	12.2 *	0.703	0.621	0.372
	(0.745–1.32)	(0.051–0.154)	(0.713–2.528)	(5.57–18.9)	(0.562–0.908)	(0.482–0.871)	(0.198–0.647)
Kékem vs. KIS	3.05	5.69 *	11.2 *	2.04	0.646	1.37	1.15
	(0.957–7.09)	(4.35–8.20)	(4.73–40.7)	(0.821–4.04)	(0.443–1.20)	(0.990–1.73)	(0.666–3.02)
Kékem vs. NG	0.802	3.10 *	4.28 *	1.37	0.463	0.642	0.761
	(0.313–1.67)	(1.92–4.13)	(2.61–6.86)	(0.658–2.28)	(0.331–0.712)	(0.475–0.830)	(0.603–0.974)
Tibati vs. KIS	2.25	0.332	1.69	0.902	1.025	1.95	2.42 *
	(1.15–3.57)	(0.191–0.549)	(0.713–5.87)	(0.519–1.39)	(0.771–1.73)	(1.48–2.52)	(1.27–7.47)
Tibati vs. NG	0.593	0.180	0.650 (0.408–0.945)	0.607	0.735	0.914	1.61 *
	(0.455–0.840)	(0.094–0.271)	(0.408–0.945)	(0.416–0.786)	(0.567–1.01)	(0.701–1.20)	(1.16–2.61)
Bélabo vs. KIS	7.47 *	3.18	7.37	3.49 *	1.48	1.54	1.93
	(3.83–10.9)	(2.10–5.38)	(1.36–40.9)	(2.50–5.43)	(0.969–2.83)	(1.21–1.99)	(0.980–4.94)
Bélabo vs. NG	1.97 *	1.73	2.83	2.35 *	1.06	0.725	1.28
	(1.40–2.61)	(0.945–2.82)	(0.780–6.91)	(1.35–3.51)	(0.725–1.69)	(0.582–0.946)	(0.994–1.53)
Santchou vs. KIS	0.641	0.934	5.27	0.848	2.17 *	0.814	1.29
	(0.286–1.25)	(0.586–1.77)	(1.31–24.1)	(0.324–1.73)	(1.44–4.10)	(0.674–0.959)	(0.810–3.14)
Santchou vs. NG	0.169	0.508	2.02	0.570	1.557 *	0.382	0.854
	(0.108–0.295)	(0.262–0.930)	(0.754–3.94)	(0.260–0.981)	(1.08–2.45)	(0.323–0.460)	(0.750–1.10)

KIS: Kisumu susceptible laboratory strain; NG: Ngousso susceptible laboratory strain.

## Data Availability

All the data from the study are included in the manuscript.

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
