# Peer review of "Pyrethroid Resistance Situation across Different Eco-Epidemiological Settings in Cameroon"

_molecules, 2022, doi:10.3390/molecules27196343_

Round 1
Reviewer 1 Report
The manuscript illustrates a primarily descriptive study on the profile of insecticide resistance in different eco-epidemiological settings across Cameroon.
The level of originality is not particularly significant as it is part of a context of similarly cited works. Nevertheless, the continuous monitoring of the spread of resistance in malaria vectors in enedemic countries is of fundamental importance for the implementation of insecticide resistance management strategies in different regions. Therefore the manuscript has an indubitable epidemiological value.
The experiental design is claer and appropriate to the proposed objectives, the organisation of the manuscript is well done. The conclusions are coherent with the results.
Minor flaws:
Line 36: all the genes should be in Italics
Line 87: The sentence starts with a citaion (n. 29) and then likely all the reference should be carefully checked.
Author Response
Reviewer 1 comment
Comment : Line 36: all the genes should be in Italics
Response : all genes have been written in italics.
Line 87: The sentence starts with a citation (n. 29) and then likely all the reference should be carefully checked.
Response : all references were checked.
Reviewer 2 Report
The article “Pyrethroid resistance situation across different eco-epidemiological settings in Cameroon”
by Armanda gives an interesting update on the pyrethroid resistance situation in malaria vector species from Cameroon.
An important sampling effort and a rich and complete set of analysis has been performed in order to evaluate levels of phenotypic resistance observed in field populations and the possible underlying causes.
Overall, the article is clearly and well written, gives good background information, clear information on the used methods and obtained results but I would appreciate a more elaborate discussion, trying to compare data with overall information available for Anopheles mosquitoes not only in Cameroon, and trying to put together information obtained with the different assays performed. Overall I would suggest minor revisions to be performed.
Minor comments:
Line 48-52: I would suggest to add also the information on how important pyrethroids are for IRS and LLIN usage since they are the only insecticides recommended for these applications.
Line 63-66: please reformulate; I would suggest something like:” Aminoacid substitutions which lead to resistance to pyrethroids or DDT are commonly referred to as knock-down resistance (kdr) mutations. The most well-known involving the replacement….”
Line 67: due to the substitution of asparagine by tyrosine
Line 69: at the end of the paragraph on target site resistance I would include also a citation of the paper by Clarkson et al 2021 “The genetic architecture of target-site resistance to pyrethroid insecticides in the African malaria vectors Anopheles gambiae and Anopheles coluzzii” which examines nicely the possible existence of other target site mutations.
Line 80: of the pyrethroid resistance situation
Line 82: profiles
Line 120-125: mosquitoes were stored differently, I guess to allow different follow-up analysis. Could you please state somewhere which analysis was performed on which type of samples? As far as I understood you performed analysis of expression levels on 30-40 specimens which were not exposed to insecticides. Did you genotype specimens exposed to insecticides for something, for example for species ID or for kdr genotypes? Was there any significant correlation between test outcomes and molecular analysis? This information should be given also in the results section.
Line 179: Table S2 is missing.
Line 180: could you please include also a table in supplementary materials where you give details on bioassay results along with the number of exposed specimens /assay?
Line 194 and 209: when referring to resistance intensity and synergist assays: results referring to 1x concentration and to exposure without PBO are the same as the ones shown in figure 2 or where they repeated? If they are the same as in figure 2, in the case of more than one bioassay performed for a sampling sites (as for example for site Bertoua which has been tested three times) which data has been used for the comparison? Also in this case a supplementary table with more detailed results would be helpful.
Iine 223: did you observe any linkage between 1014F and 1575Y allele ( as observed previously by Jones et al 2012).
Line 235 and 237: how do you define high and moderate overexpression?
Line 239: Could you please explain better why you name these four enzymes? From the table for example Cyp4G16 does not seem to have clearly higher fold changes.
Line 265 : profiles
Line 270: patterns
Line 302: could you discuss this point further? Where are data on An.gambiae and An.coluzzii?
Line 303: delete kdr
Line 311 and 329: the observation of full recovery with the synergist assay and the higher level of CHCs in in Kaele is interesting also since this is the only village with a slightly lower frequency of the 1014F mutation
Line 340-343: could you add some more information on how these results can be used in vector control programs.
Tables and Figures: please revise legends and give more complete information.
Table 2: I do not see any bold letters as described in the legend
Table S2 is missing.
Author Response
Reviewer 2 comment
We would like to thank the reviewer for his comments which substantially improved the quality of the manuscript
Minor comments:
Comment: Line 48-52: I would suggest to add also the information on how important pyrethroids are for IRS and LLIN usage since they are the only insecticides recommended for these applications.
Response: We thank the reviewer for this comment the sentence was revised accordingly
Comment: Line 63-66: please reformulate; I would suggest something like:” Aminoacid substitutions which lead to resistance to pyrethroids or DDT are commonly referred to as knock-down resistance (kdr) mutations. The most well-known involving the replacement….”
Response: We thank the reviewer for this comment the sentence was revised accordingly
Comment: Line 67: due to the substitution of asparagine by tyrosine
Response: We thank the reviewer for this comment the sentence was revised accordingly
Comment: Line 69: at the end of the paragraph on target site resistance I would include also a citation of the paper by Clarkson et al 2021 “The genetic architecture of target-site resistance to pyrethroid insecticides in the African malaria vectors Anopheles gambiae and Anopheles coluzzii” which examines nicely the possible existence of other target site mutations.
Response: The citation was included in the manuscript as recommended
Comment: Line 80: of the pyrethroid resistance situation
Response: corrected
Comment: Line 82: profiles
Response: corrected
Comment: Line 120-125: mosquitoes were stored differently, I guess to allow different follow-up analysis. Could you please state somewhere which analysis was performed on which type of samples? As far as I understood you performed analysis of expression levels on 30-40 specimens which were not exposed to insecticides. Did you genotype specimens exposed to insecticides for something, for example for species ID or for kdr genotypes? Was there any significant correlation between test outcomes and molecular analysis? This information should be given also in the results section.
Response: Mosquitoes exposed to insecticide were genotyped to determine species ID, kdr genotypes and molecular mechanisms. Due to the reduce sample size we feel the information should not be accurate. We are at the moment conducting a larger study which will capture this information.
Comment: Line 179: Table S2 is missing.
Response: Sorry the Table S2 was removed before submission please consider the Figure 3A and 3B
Comment: Line 180: could you please include also a table in supplementary materials where you give details on bioassay results along with the number of exposed specimens /assay?
Response: The table was added Table S2 was added
Comment: Line 194 and 209: when referring to resistance intensity and synergist assays: results referring to 1x concentration and to exposure without PBO are the same as the ones shown in figure 2 or where they repeated? If they are the same as in figure 2, in the case of more than one bioassay performed for a sampling sites (as for example for site Bertoua which has been tested three times) which data has been used for the comparison? Also in this case a supplementary table with more detailed results would be helpful.
Response: In places where more than one test was performed we determined the mean mortality rates graph was used to plot graphs. Table S2 has been added for additional information
Comment: Line 223: did you observe any linkage between 1014F and 1575Y allele (as observed previously by Jones et al 2012).
Response: This was not look at due to the low frequency of the N1575Y mutation in our sample but a study is underway and is intended to assess this
Comment: Line 235 and 237: how do you define high and moderate overexpression?
Response: While there is no a general predefined threshold to define high and moderate overexpression, we choose to characterize statistically significant overexpression between 1.5-3.0 folds as moderate and >3.0 as high overexpression.
Comment: Line 239: Could you please explain better why you name these four enzymes? From the table for example Cyp4G16 does not seem to have clearly higher fold changes.
Response : Corrected accordingly see result section
Comment: Line 265: profiles
Response: The word was corrected in the manuscript
Comment: Line 270: patterns
Response: The word was corrected in the manuscript
Comment: Line 302: could you discuss this point further? Where are data on An.gambiae and An.coluzzii?
Response : Data on An.gambiae and An.coluzzii are represented as pie in the figure 3A and 3B a sentence was added in the result section to indicate this
Comment: Line 303: delete kdr
Response: kdr was deleted
Comment: Line 340-343: could you add some more information on how these results can be used in vector control programs.
Response: The following sentence was added « Furthermore, the implementation of integrated control strategies could be critical to address challenges in different ecoepidemiological settings and to improve malaria control in the country.».
Comment: Tables and Figures: please revise legends and give more complete information.
Table 2: I do not see any bold letters as described in the legend
Response: The table and figure legends were revised accordingly.
Comment: Table S2 is missing.
Response: A new Table S2 have been added with information on susceptibility tests done
